# Mechanical Performance of Patched Pavements with Different Patching Shapes Based on 2D and 3D Finite Element Simulations

Shujian Wang [1,2], Han Zhang [3], Cong Du [1,4,*], Zijian Wang [3], Yuan Tian [1,*] and Xinpeng Yao [3]

1  School of Qilu Transportation, Shandong University, Jinan 250061, China
2  Shandong Hi-Speed Construction Management Group Co., Ltd., Jinan 250001, China
3  Shandong Hi-Speed Group Co., Ltd., Jinan 250098, China
4  Suzhou Research Institute, Shandong University, Suzhou 215123, China
*  Correspondence: congdu@sdu.edu.cn (C.D.); yuantian@sdu.edu.cn (Y.T.)

**Abstract:** Patching is a common technology used in repairing asphalt-pavement potholes. Due to the differences in material properties between patched- and unpatched-asphalt mixtures, significant strain and stress concentrations could be induced; thus, further cracks and interfacial debonding distress could be caused. As a remedy, the strain and stress concentrations can be alleviated by utilizing optimum patching shapes. Therefore, this paper employed finite element methods (FEM) to deeply analyze the mechanical performance of patched-asphalt pavements embedded with different patching shapes. Three patching shapes, these being rectangular, stair, and trapezoid, were considered for use in pavement pothole repairs based on two- and three-dimensional finite element models. In the two-dimensional models, Top-Down and Bottom-Up crack propagations were simulated to assess the anti-damage performance of the patched pavements with different patching shapes. In addition, the thermal stress behaviors within patched-asphalt pavements were simulated using the two-dimensional model to analyze the performance of the patched pavements during the cooling process in construction. In addition, interface-debonding performance was simulated for the patched-asphalt pavements using three-dimensional models. In light of the simulation results, engineers are expected to better understand the mechanism within patched pavements and to improve the quality of the pavement patching.

**Keywords:** patching shape; finite element method (FEM); crack propagation; thermal stress; interface debonding

## 1. Introduction

Asphalt pavements are commonly subject to pothole distress induced by fractures occurring in the surface layer. Mostly, potholes are formed as when the initial distresses are not adequately repaired, and in such cases the initial distresses could be caused by various factors, including moisture damage and heavy traffic loads [1]. As a remedy, pothole patching is usually employed as a low-cost treatment and emergency repair for pothole distress to ensure the service performance of asphalt pavements [2]. Unlike high-cost pavement reconstruction, the patching method replaces the pothole area with new hot mixed asphalt (HMA) in an asphalt pavement [3]. Therefore, the workability of the patched-asphalt mixtures determines the service performance and life cycle of the asphalt pavements.

The employment of patching technology in pavements significantly contributes to the rehabilitation of asphalt pavements and further reduces the cost of pavement mainte-nance [3]. Nonetheless, the inherent properties of the original and patched-asphalt mixtures are distinct due to different aging effects. The original asphalt mixture has served over a long time, by which time the asphalt binder has aged due to the long-term oxidative effects. Therefore, the original asphalt mixture becomes stiffer and more brittle. The patched

mixture, on the other hand, has usually been produced recently or in the field. Hence, the patched mixture is only subject to the short-term aging effects, in which its properties are influenced by the heating and mixing process performed during the asphalt mixture's production [4,5]. Consequently, the original asphalt mixtures have higher stiffness and lower fracture resistance than the patched ones. Hence, remarkable stress/strain concentrations would be induced near the interface between the original and patched-asphalt mixtures in pavements on account of the different mechanical properties in the two mixtures.

To solve this problem, numerous studies have been made to promote the workability of the patched mixtures by manipulating the mechanical and chemical properties of materials. To guarantee the continuity of stress and deformations in patched pavements, the mechanical performance of patching materials should be similar to the original asphalt mixture. In addition, the chemical properties of patched and original materials determine the adhesion between patched and unpatched parts. For example, Kwon, et al. [6] analyzed the performances of patching mixtures made with recycled materials and compared them to the implementation of mixtures made with virgin materials. The results showed that the mechanical behavior of patched mixtures depended on the types of constituents used. The mixtures containing recycled materials performed with better stabilities and adhesion properties than those made with virgin materials. Yang, et al. [7] reported a patching material produced by an epoxy-asphalt binder and fine gradation, which can be used in repairing the potholes of deck pavements over steel bridges. The fatigue properties and viscoelastic responses of the proposed patching materials were evaluated by conducting the three-point bending fatigue tests and numerical analyses. In addition, the cold-patched-asphalt material was developed due to its high performance and is less likely to be affected by seasons [8]. Liu, et al. [9] provided detailed descriptions of the material composition, compaction process, and repair procedure of patched pavements using cold-mix asphalt. Yuan, et al. [10] improved the moisture stability of a cold-patched-asphalt mixture by incorporating cement, and the performance of the mixture with different cement contents was tested by Marshall tests and freeze–thaw indirect tensile tests. Resulting from the inherent properties of the patched mixture, the interfacial debonding performance showed primary distress between patched- and unpatched-asphalt mixtures. The adhesion between patched and original materials determines the interfacial performance of the patched pavement, which is not only controlled by the chemical and mechanical properties of materials but is also affected by the shapes and sizes of the interfaces. To address this issue, many efforts have been made towards improving the interfacial adhesion between patched- and unpatched-asphalt mixtures. Li, Huang, Shao, and Ren [1] conducted orthogonal experiments to assess the service lives of pothole repairs with various interface shapes, and they determined the optimum interface shape of patching by evaluating the joint effects in the interface. Eisa, Abdelhaleem, and Khater [3] employed experimental approaches to investigate the interfacial fractures of a patched-asphalt mixture with steel-fiber reinforcements under various loading conditions, and the results indicated that the patching shape had more impact on the interface-debonding performance than the patching depth. Byzyka et al. [11,12] investigated temperature distributions at interfaces between patched and unpatched mixtures during the construction of asphalt-pavement-patch repairs, and they suggested that a pre-heating procedure in pothole repair is necessary to increase the workability of the patching-asphalt mixtures.

The abovementioned studies effectively optimized the asphalt-pavement-pothole-patching-repair technology by modifying the materials and construction manners of the patched-asphalt mixture. However, the influences of the patching shapes on the mechanical behavior of the repaired asphalt pavement were seldom reported. In reality, potholes usually have irregular shapes and sizes. To improve repairs and guarantee the performance of the patched pavements, the pothole areas are cut with a saw into regular shapes in engineering practice. The shape of pothole patching determines the interfaces between patched- and unpatched-asphalt mixtures, and further affects the critical stress–strain performance of repaired asphalt pavements. As mentioned above, the material properties

of patched- and unpatched-asphalt mixtures are different, and such differences in asphalt pavements could induce significant strain/strain concentrations. Consequently, crack damages can easily propagate near the interface between patched- and unpatched-asphalt mixtures. Therefore, a deep insight into the mechanics of asphalt pavements embedded with different patching shapes is necessary for the design and construction of asphalt-pavement repair.

To this end, the finite element method (FEM) is employed in this research to analyze the stress–strain behavior of asphalt pavements with different asphalt-mixture-patching shapes. The FEM is a numerical tool widely used in characterizing the detailed mechanical performance of structures by discretizing the structures into small and simple elements. By solving the partial differential equations of those small elements and connecting them by common nodes, the complex mechanical responses can be effectively illustrated in contour plots. Therefore, the FEM is widely accepted as an effective tool in pavement design for analyses without exhausting labor work and high cost [13–16]. Furthermore, the extended finite element method (XFEM) was employed to investigate crack propagation in asphalt pavements. The XFEM defines an enriched finite element area and initial crack in a structure, and enables a crack to be propagated based on the critical stress responses. Based on XFEM technology, scholars analyzed the crack behaviors of asphalt pavements subjected to various loading conditions, including Hossain, et al. [17], Wang, et al. [18,19] and so on.

In this study, three shapes were considered in the asphalt-pavement patching, which were rectangular, stair, and trapezoid shapes. The internal mechanical performance of asphalt pavement embedded with the three patches were investigated by FEM. To concentrate on the crack propagations throughout the patched- and unpatched-asphalt mixtures, two-dimensional finite element models of the three patched-asphalt pavements were developed using ABAQUS/CAE 2017 software, and the interfaces between patched and unpatched parts were specified as perfectly bonded. Pre-defined initial cracks were specified in the surface and bottom of the asphalt layer and XFEM was employed to simulate the Top-Down and Bottom-Up crack propagations, respectively. In addition, crack propagations in the unpatched pavements were simulated for comparison purposes. Furthermore, the thermal-stress distributions within patched-asphalt pavements were simulated using the two-dimensional model to analyze the mechanical performance of the patched pavements during the cooling process in construction. In addition, the interface-debonding performances of patched-asphalt mixtures with respect to different patching shapes were simulated using three-dimensional models, and different tire-loading locations were taken into account. The results can serve as important references for the future design and construction of asphalt-pavement patching.

## 2. Methodology

### 2.1. The Extended Finite Element Method (XFEM)

For modeling stationary discontinuities, such as a crack, the extended finite element method (XFEM) is commonly employed in reducing the mesh refinements near the crack tips [20–25]. The XFEM is the extension of the regular FEM, in which the local enrichment functions are incorporated to ensure discontinuities without considerable mesh refinements in FE models.

In the XFEM analyses, the singularity around the crack tip and the jump across the crack surfaces are represented by the near-tip asymptotic function and the discontinuous function, respectively, within the enriched elements. Therefore, the displacement associated with the crack propagation can be approximated as

$$\boldsymbol{u} = \sum_{I=1}^{N} N_I(x) \left[ \boldsymbol{u}_I + H(x)\boldsymbol{a}_I + \sum_{\alpha=1}^{4} F_\alpha(x)\boldsymbol{b}_I^\alpha \right] \qquad (1)$$

where $u$ represent the displacement vector function; $N_I(x)$ are the functions relating to the shape of the element; the three terms on the right-hand side refer to the continuous displacements in the regular elements ($u_I$), the jump in displacement across the crack surfaces ($H(x)a_I$) and the discontinuous displacements around the crack tip ($\sum_{\alpha=1}^{4} F_\alpha(x)b_I^\alpha$), respectively; $H(x)$ and $F_\alpha(x)$ are the jump function and asymptotic crack-tip function associated with crack interior and crack tip, respectively; and $a_I$ and $b_I^\alpha$ are the nodal-enriched degree of freedom vectors corresponding to the abovementioned two functions.

During the crack propagation, the elements relating to the crack path are enriched, as presented in Figure 1. The three terms on the right-hand side in Equation (1) are used with respect to different situations. The $u_I$ is applicable to all the elements; the $H(x)a_I$ is used for elements cut by the crack interior; and the $\sum_{\alpha=1}^{4} F_\alpha(x)b_I^\alpha$ is only valid for elements cut by the crack tip. The jump function and asymptotic crack-tip function are presented as Equations (2) and (3), respectively,

$$H(x) = \begin{cases} 1 & (x - x^*) \cdot n \geq 0 \\ -1 & \text{otherwise} \end{cases} \tag{2}$$

$$F_\alpha(x) = \left[ \sqrt{r}\sin\frac{\theta}{2}, \ \sqrt{r}\cos\frac{\theta}{2}, \ \sqrt{r}\sin\theta\sin\frac{\theta}{2}, \ \sqrt{r}\sin\theta\cos\frac{\theta}{2} \right] \tag{3}$$

where $x$, $x^*$ are the Gauss point and its closest foot point to the crack path; $n$ is the unit outward-normal to the crack at $x^*$; and $(r, \theta)$ is the polar coordinate system with the origin at the crack tip.

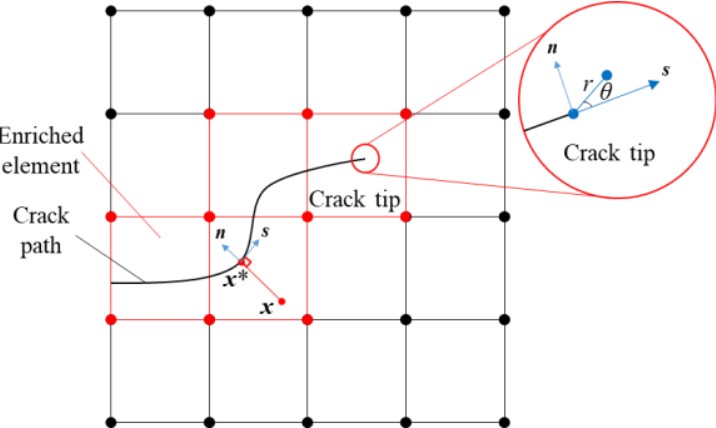

**Figure 1.** Enriched elements in XFEM.

To define the crack initiation and propagation, the maximum stress principle is utilized, and the crack will propagate when the following equation is satisfied:

$$\left\{ \frac{\langle \sigma_{\max} \rangle}{\sigma_{\max}^0} \right\} = 1 \tag{4}$$

where $\sigma_{\max}^0$ stands for the strength of the structures and the symbol $\langle \rangle$ represents the Macaulay bracket.

Once the maximum stress principle is reached, the crack will propagate in the enriched elements. Correspondingly, a scalar damage variable denoted $D$ will increase from 0 to 1, and the function of $D$ is formulated as

$$D = \frac{\delta_{\mathrm{m}}\left(\delta_{\mathrm{m}}^{\max} - \delta_{\mathrm{m}}^0\right)}{\delta_{\mathrm{m}}^{\max}\left(\delta_{\mathrm{m}} - \delta_{\mathrm{m}}^0\right)} \tag{5}$$

where $\delta_{\mathrm{m}}^{\max}$ represents the maximum displacement; $\delta_{\mathrm{m}}$ is the effective displacement at failure; and $\delta_{\mathrm{m}}^0$ is the effective displacement at damage initiation.

*2.2. Model Development*

To effectively investigate the crack propagations in the entire surface layer in pavements, the two-dimensional asphalt-pavement model with enriched elements was established based on the FE program ABAQUS, and the pavement layer structure was specified according to specifications [26], as shown in Figure 2. To characterize the realistic performance of asphalt pavements, the 4-node bilinear plane strain quadrilateral elements (CPE4) were employed in the model. Moreover, the mesh size for the surface layer and the patched-asphalt mixture was defined as small enough (approximately 1 cm) to guarantee the simulation convergence. In this study, three types of patching shapes were defined: a rectangular shape, stair shape, and trapezoid shape, respectively. In reality, the sizes of patched areas are usually larger than the tire-loading areas. However, too large a size of the patch would generate numerous elements. Thus, the simulation efficiency would be reduced, and the mechanical details would be lost. Therefore, the size of the patch area in this study was defined as 30 cm in width, which was slightly larger than the tire-loading areas (20 cm), as presented in Figure 3. In addition, the trapezoidal shape has 45-degree-sloped sides to reduce the modeling efforts. The XFEM technology was employed to investigate the crack propagation in asphalt pavement. To demonstrate the Top-Down and Bottom-Up crack performances of asphalt pavements, pre-defined cracks were inserted in the surface and bottom parts of the asphalt layer, respectively. Moreover, the interfaces between patched and unpatched parts in pavements were specified as perfectly bonded to allow the cracks to propagate throughout the entire surface layers. In the Top-Down cracking simulations, the initial cracks were inserted near the edge of the patched part in the top area of the asphalt layer. According to [27], the maximum tensile stress was induced at some distance away from the tire edge location within asphalt pavements. Therefore, the tire load was put on the patched-asphalt mixture, and the distance between the loading area and the initial crack was 5 cm, as shown in Figure 3. In the Bottom-Up cracking simulation, the initial cracks were inserted near the interface between the patched- and unpatched-asphalt mixtures. The tire loading was applied above the crack locations, and the distance between the crack location and the tire edge was 5 cm, as given in Figure 3. The initial crack in the simulation represented the internal flaws in asphalt pavements, including air voids and micro cracks. Therefore, the sizes of the initial cracks were specified as being small enough. The length of each crack was specified as 0.5 cm, which is smaller than the mesh size (1 cm). In addition, the initial cracks were specified as having zero width, and hence the impact of the cracks' width on the simulation results was neglected.

The three-dimensional FE model of the patched pavement was developed to investigate the interface-debonding performance of the three patched structures. The layered structures of the three-dimensional model were the same as the two-dimensional model. in contrast, the interfaces between patched and unpatched parts were defined as bonded with finite sliding. Therefore, the interface-debonding performance of the patched pavements can be effectively demonstrated by the relative displacements in the interfaces. The three-dimensional model of asphalt pavements and three patched parts are provided in Figure 4. According to the features of pavement structure, the continuum three-dimensional 8 node solid elements (C3D8) were employed on the three-dimensional models [28]. The mesh size for the three-dimensional model were defined as 5 cm to balance the computational efforts and simulation convergence.

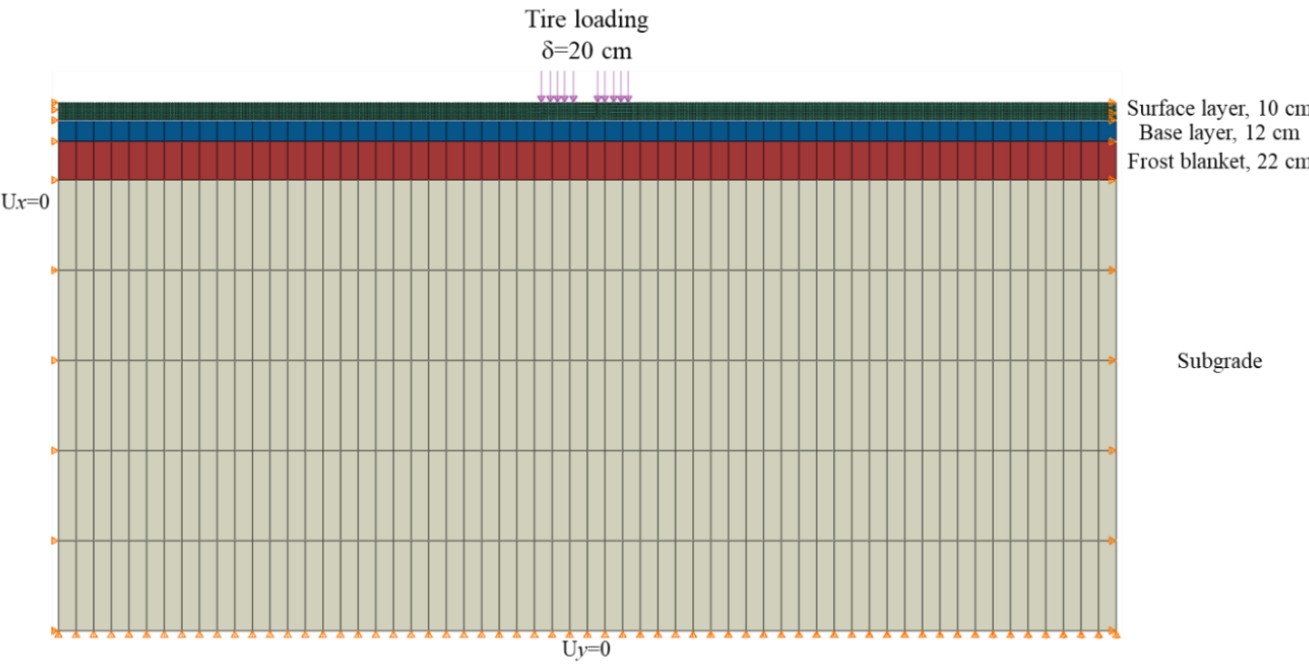

**Figure 2.** Dimensional model of asphalt pavement.

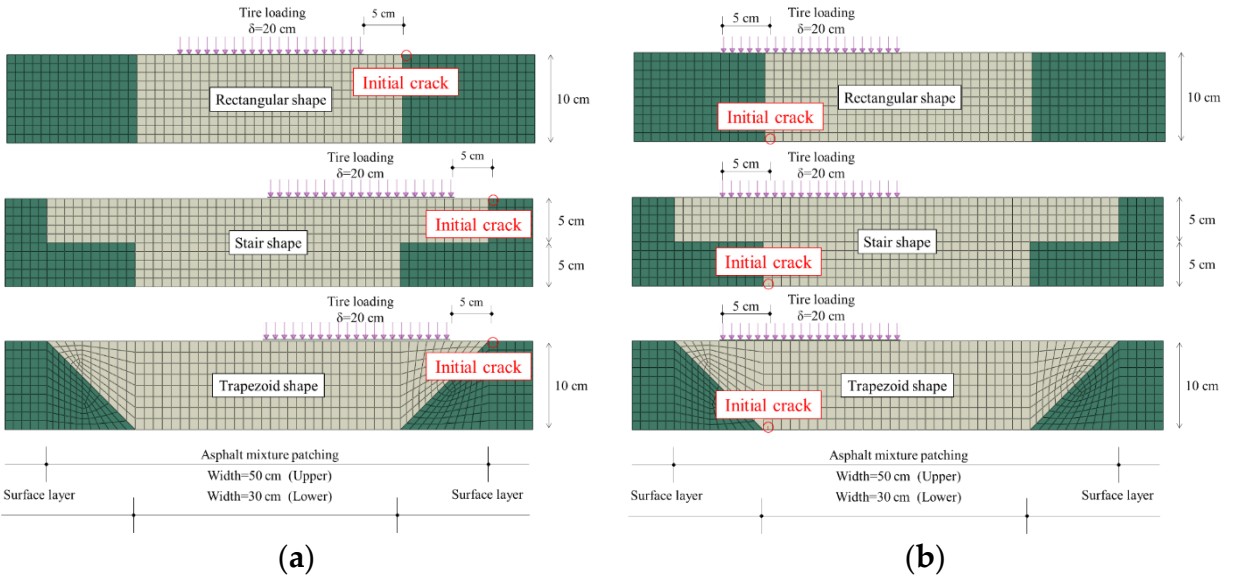

**Figure 3.** Simulations of the patched-asphalt mixture for (**a**) Top-Down crack propagation and (**b**) Bottom-Up crack propagation.

*2.3. Model Parameters*

Asphaltic materials are commonly characterized using linear viscoelasticity [28], which performs time-dependent and temperature-dependent mechanical behaviors. However, in this study the loading frequencies and temperatures for the abovementioned models were identical; therefore, the simulation results can be focused on the effects of different patching shapes. To simplify the simulation, the linear elastic characteristic was employed on the patched and unpatched asphalt-pavement layers. Referring to [29], the material coefficients for pavement layers were specified, as presented in Table 1. Moreover, the fracture properties, strength, and fracture energy, were defined according to [19]. In engineering, the patched parts of asphalt pavements are generally made from new asphalt mixtures, while the unpatched asphalt mixture has aged during long-term service. Thus,

the mechanical properties of the patched and unpatched parts in asphalt pavement can be specified in terms of the short-term- and long-term-aged asphalt concrete, respectively. According to [4,5], the long-term-aged asphaltic materials have higher stiffness and lower fracture resistance than the short-term-aged ones. Thus, according to the parameters of the asphalt layer in the pavement, the Young's modulus, tensile strength, and fracture energy of the patched-asphalt mixture were specified as 20% reduction, 10% reduction, and 10% increase relative to the unpatched surface layer, respectively [5]. All the model parameters are given in Table 1.

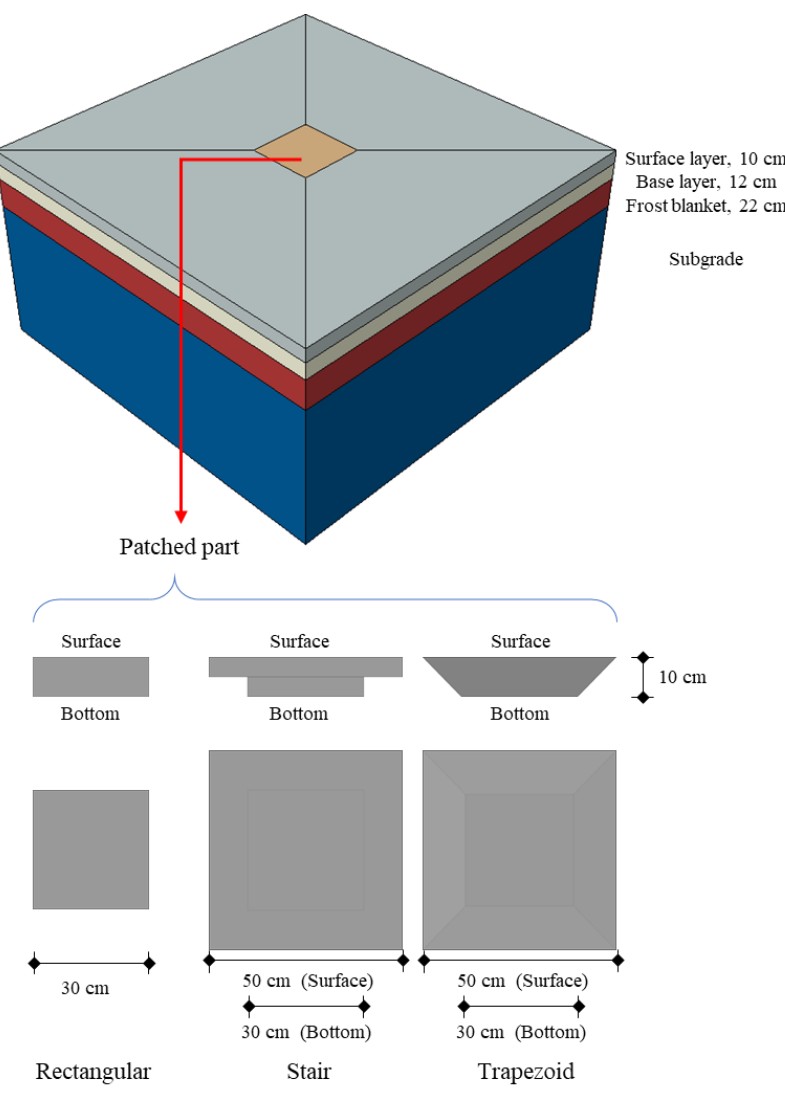

**Figure 4.** Dimensional models of asphalt pavement and patched parts.

**Table 1.** Model parameters.

|  | Modulus (MPa) | Poisson's Ratio | Tensile Strength (MPa) | Fracture Energy (J/m²) |
|---|---|---|---|---|
| Surface layer | 11,150 | 0.35 | 2 | 1.5 |
| Asphalt-mixture patching | 8920 | 0.35 | 1.8 | 1.67 |
| Base layer | 6893 | 0.25 | - |  |
| Frost blanket | 125.7 | 0.45 | - |  |
| Subgrade | 98.9 | 0.45 | - |  |

*2.4. Simulation*

In the two-dimensional XFEM simulation, static tire loadings were applied on patched and unpatched asphalt pavements with pre-defined initial cracks, and the loading value was 0.7 MPa. To deeply investigate the effects of the patched part on the overall mechanical performance of asphalt pavements, initial cracks were inserted near the interface between the original- and patched-asphalt mixtures, as shown in Figure 3. By specifying the locations of tire loading and initial cracks, the Top-Down crack propagation and Bottom-Up crack propagation can be effectively simulated. In addition, the temperature-declining process of the patched-asphalt mixture was simulated. According to engineering practices, the temperature in the production of asphalt mixtures is over 150 °C [30], and the temperature in the asphalt-pavement surface layer is about 30 °C [31]. While the temperature is cooling from 150 °C to 30 °C, the patched-asphalt mixture would be contracting due to the volume reduction. However, the patched parts were bonded by the unpatched asphalt pavements, and hence the thermal stress could be induced in asphalt pavement due to the contraction of the patched part cooling from 150 °C to 30 °C. The thermal stress within the asphalt mixture during the patching process was also simulated using the two-dimensional finite element model, in which the expansion coefficient was $2 \times 10^{-5}$ [31], and horizontal- and shear-stress behaviors were discussed.

In two-dimensional simulations, the interface between patched and unpatched parts in the pavement was perfectly bonded to enable focus on the crack performance in the surface layer. However, debonding performance can be induced in the interface due to the differences in material properties between patched- and unpatched-asphalt mixtures. Such interface-debonding performance relates to the locations of tire loadings and complex three-dimensional structures. Therefore, the three-dimensional models of the rectangular-patched, stair-patched, and trapezoid-patched pavements were developed, and different tire-loading locations were applied to simulate the interfacial behavior. As presented in Figure 5, three configurations were defined to represent three loading locations, which were the center, edge, and corner of the patched parts, respectively. The loading was applied on the circle area with diameter of 20 cm with a value of 0.7 MPa.

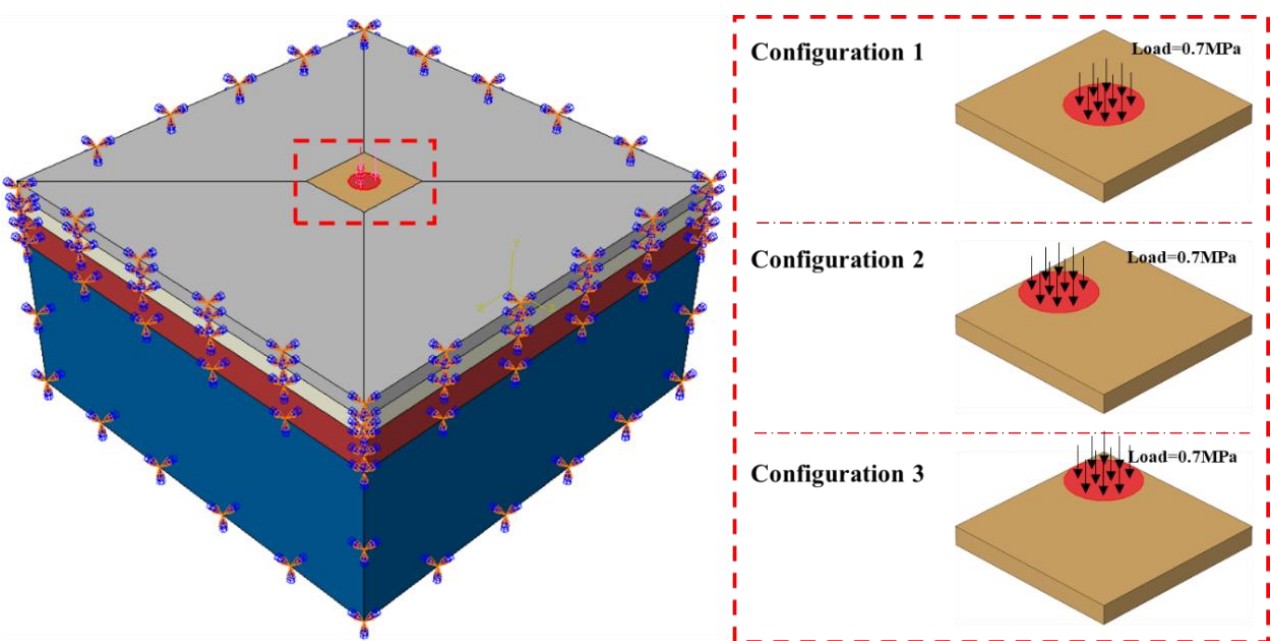

**Figure 5.** Loading configurations in three-dimensional simulations.

## 3. Results and Discussion

### 3.1. Top-Down Crack Propagation

As presented in Figure 3a, the Top-Down cracking process was simulated on the two-dimensional asphalt-pavement model with an initial crack inserted near the pavement's surface. Consequently, the scalar damage variables (*D*) and strain distributions were extracted to exhibit the mechanics of the Top-Down crack propagation within asphalt pavements with different patched shapes, as shown in Figures 6 and 7. As mentioned in Equation (5), the scalar damage variable (*D*) represents the degradation of the stiffness once the corresponding initiation criterion has been reached. While *D* is increasing from 0 to 1, the stiffness of the corresponding elements is decreasing from nondamaged to zero. In addition, the unpatched asphalt pavement was simulated for comparison.

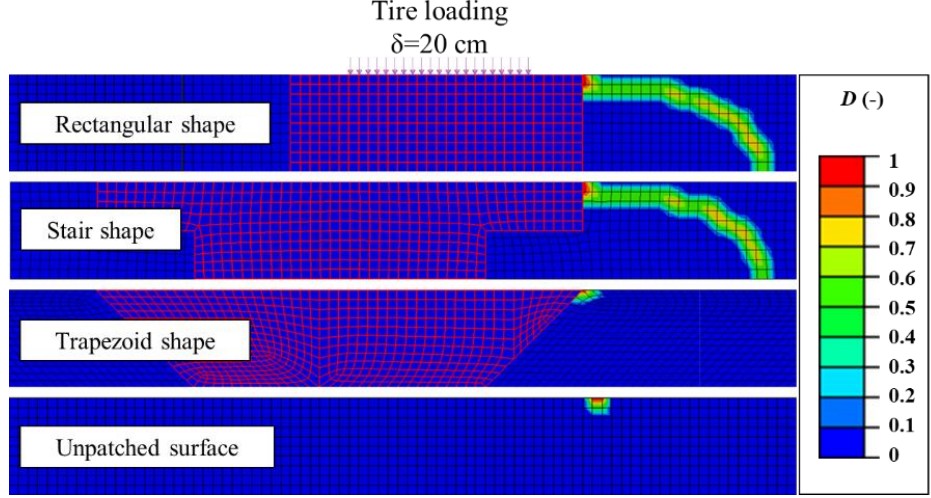

**Figure 6.** Top-Down crack simulations.

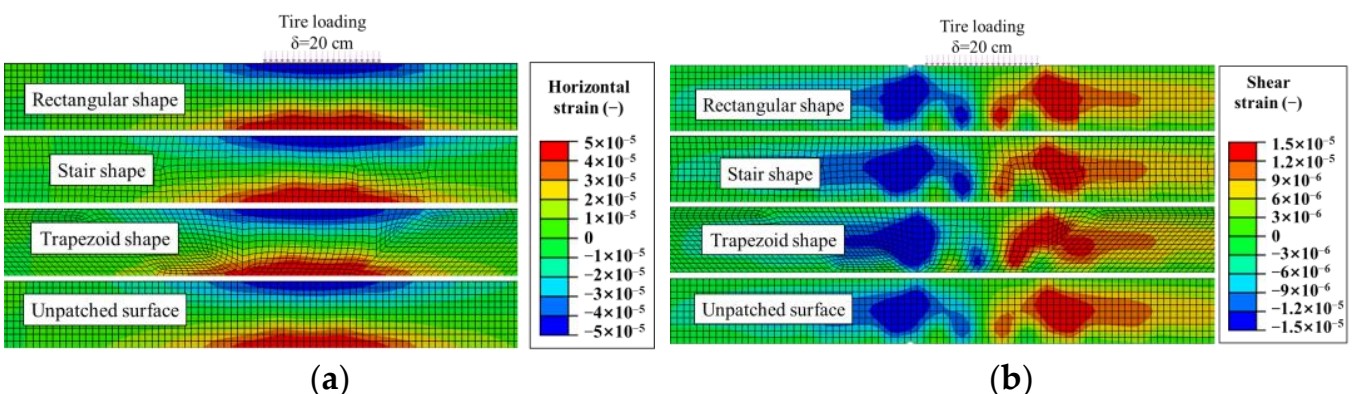

**Figure 7.** Distributions of (**a**) horizontal strain and (**b**) shear strain.

It can be observed from Figure 6 that the Top-Down crack propagations in the four different surface layers behave with remarkable differences. Serious cracks were caused in the patched-asphalt pavements with rectangular and stair shapes, while the pavement with a trapezoid-shaped patch and unpatched pavement showed no crack propagation. Hence, it can be speculated that the trapezoid shape is to be recommended for reducing the Top-Down crack propagations in asphalt pavement with patched parts according to the current results. To deeply investigate the internal mechanism of cracking performance in different pavements, the horizontal and shear strain related to the crack locations were extracted and analyzed.

In the horizontal-strain results, the strain discontinuity in the interface between patched and unpatched parts can be observed in the pavements with rectangular and stair patches. In particular, the strain difference between the rectangular patch and the pavement was more significant than that for the stair-patched pavement. Therefore, the mechanism of Top-Down crack propagation in the four pavements can be explained by the following reasons. In the rectangular- and stair-patched pavements, the patched parts were relatively "softer" due to having lower stiffness than the unpatched part, and hence the horizontal deformations between patched and unpatched parts were different. Nevertheless, the vertical interface between patched and unpatched parts cannot effectively transfer the strain and deformation between the two parts, and the initial cracks propagated as the deformation discontinuity increased. On the other hand, the trapezoid-patched-asphalt pavement had better abilities in transferring the horizontal strain and deformation by the slant interface between patched and unpatched parts, and this further reduced the strain and deformation discontinuities in pavement. As a result, the horizontal-strain behavior in trapezoid-patched pavement was close to that in the unpatched pavement. Near the surface of the pavement, the trapezoid-patched pavement and unpatched pavement both performed compressive strain states, which prevents the propagation of cracks.

Similar phenomena were also exhibited in the shea- strain behavior. The shear-strain distributions in trapezoid-patched pavement and unpatched pavement were similar to each other, while the rectangular- and stair-patched pavements showed strain discontinuities. The lower stiffness of the patched part and the vertical interface between patched and unpatched parts caused discontinuous shear strain in the rectangular- and stair-patched pavements, further accelerating the crack propagation in pavements.

According to the above results, it can be concluded that the shape of the patch parts in asphalt pavements significantly influences Top-Down crack performance. The trapezoid-patched pavement has better anti-crack performance, followed by the stair-patched pavement, and the rectangular-patched pavement is more vulnerable to Top-Down crack propagation.

### 3.2. Bottom-Up Crack Propagation

As presented in Figure 3b, the Bottom-Up cracking process was simulated on the two-dimensional asphalt-pavement model with an initial crack inserted near the bottom of the pavement. Similarly, the scalar damage variables (*D*) and strain distributions were extracted to exhibit the mechanics of the Bottom-Up crack propagation in asphalt pavements, as shown in Figures 8 and 9. In addition, the unpatched asphalt pavement was simulated for comparison consideration.

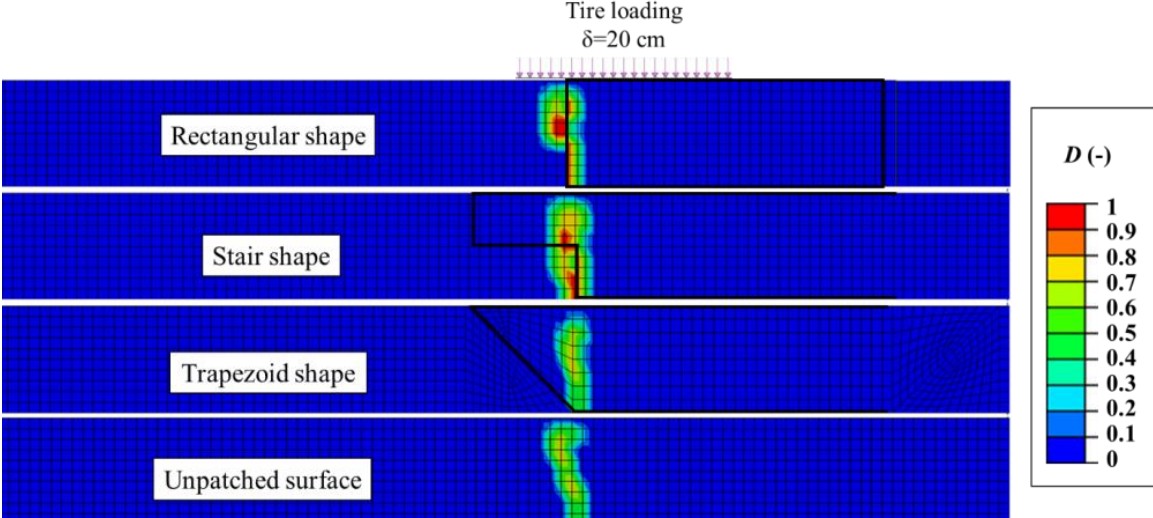

**Figure 8.** Bottom-Up crack simulations.

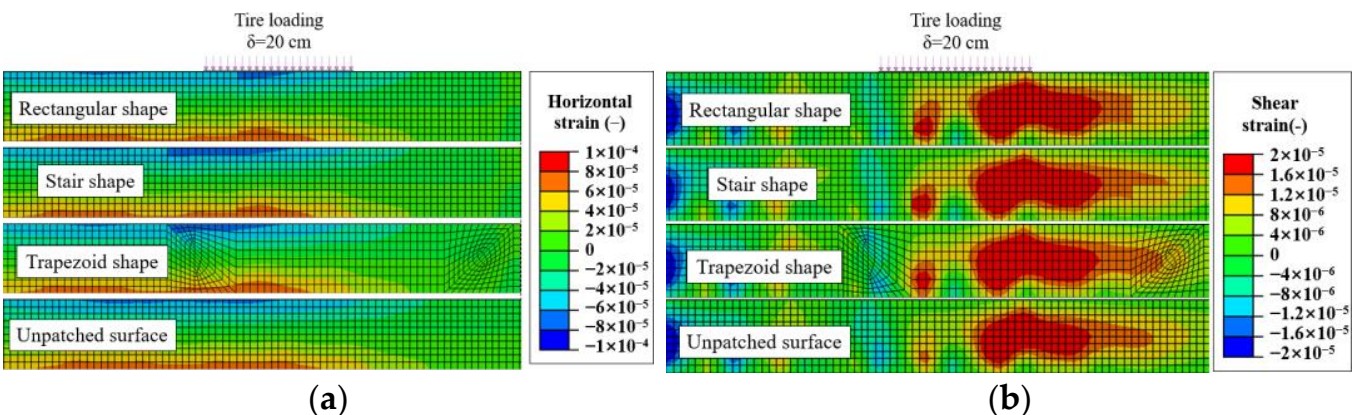

**Figure 9.** Distributions of (**a**) horizontal strain and (**b**) shear strain.

It can be seen from Figure 8 that all four pavements performed Bottom-Up crack propagations, which can be explained by the horizontal tensile strain in the bottom area, as presented in Figure 9. Severe cracks can be observed in the rectangular- and stair-patched pavements, which is in accordance with the Top-Down crack behavior. Relatively, the trapezoid-patched and unpatched pavements have slight crack propagations. Further, it should be noted that the crack in the trapezoid-patched pavement was smaller than that in the unpatched pavement, which can be ascribed to the lower fracture energy of the unpatched asphalt mixture. More information can be found in the strain distributions for the pavement.

In the horizontal-strain distributions, the lower stiffness in the patched mixture generated higher strains, and strain discontinuities can be seen in the interface between patched and unpatched mixture parts. Furthermore, the strain discontinuities in rectangular- and stair-patched pavements seem more evident due to the vertical interfaces, and consequently induced severer cracks. On the other hand, despite the higher horizontal strains in the trapezoid-patched part, the strain distributions near the interface seem smooth. Therefore, the crack propagation in the trapezoid-patched pavement was alleviated. Comparing the trapezoid-patched pavement and unpatched pavement, the larger crack was induced in the unpatched pavement despite the lower strain distributions. Such phenomenon can be explained by the different fracture resistance of patched- and unpatched-asphalt mixture. In the trapezoid-patched and unpatched pavements, the cracks propagated in the patched and unpatched parts, respectively. According to Table 1, the patched-asphalt mixture has lower strength and higher fracture energy compared with the unpatched asphalt mixture, and therefore, the fracture resistance in the patched part was relatively higher than that in the unpatched part.

To evaluate the further crack evolution in the four pavement types, the horizontal stress in the crack paths was extracted and exhibited in Figure 10. The stress distributions indicate the potential crack propagations in the pavements in the future loading conditions. The results showed significant differences with the abovementioned crack behaviors. The highest stress is situated in the bottom area of stair-patched pavement, which indicates that the crack can hardly propagate to the surface of pavements. Followed by the trapezoid-patched pavement, the next highest level of horizontal stress was distributed in the crack tip, which suggests further crack evolution from the bottom to the surface of asphalt pavements. The higher stress values in the rectangular-patched pavement were also near the crack tips, which makes the pavement vulnerable to further crack propagation. For the unpatched pavement, the tensile stress was the smallest and was mostly distributed near the crack tip, which suggests that the crack in the unpatched pavement would barely be propagated.

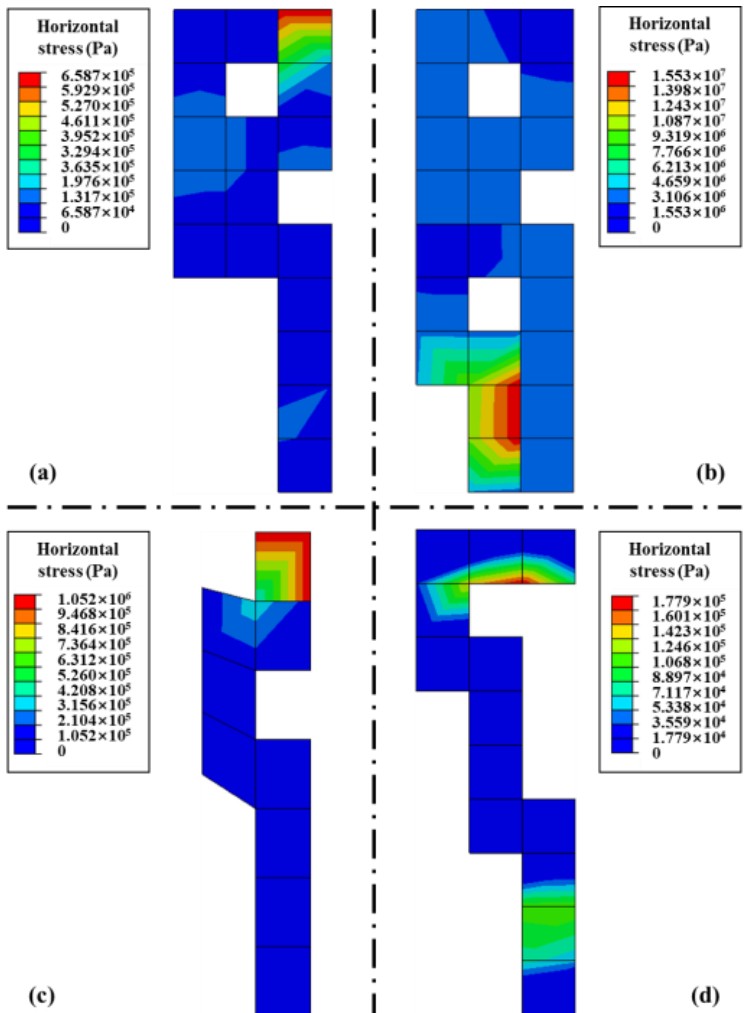

**Figure 10.** Distributions of horizontal stress at crack paths in (**a**) rectangular-patched, (**b**) stair-patched, (**c**) trapezoid-patched, and (**d**) unpatched pavements.

From the above results, it can be concluded that the patch shapes have different influences on Bottom-Up crack propagations. The trapezoid-patched pavement has the smallest crack propagation among the four pavements. However, the horizontal stress in the crack paths indicates that the trapezoid- and rectangular-patched pavements have higher crack potential than other pavements.

### 3.3. Temperature Stress

The horizontal- and shear-stress distributions in the patched-asphalt pavements during the cooling process from 150 °C to 30 °C are exhibited in Figures 11 and 12, respectively. The stress in the rectangular-patched pavement was lower than that in the other two patched pavements, which might be because the volume of the rectangular patching was the smallest among the three patched shapes. Therefore, lower stress was caused in the rectangular-patched pavement during the cooling-down process. For the other two patched pavements, the horizontal-stress distributions were similar to each other in the stair- and trapezoid-patched-asphalt pavements; however, the shear stress in the trapezoid-patched pavement was much higher than that in the stair-patched asphalt pavement.

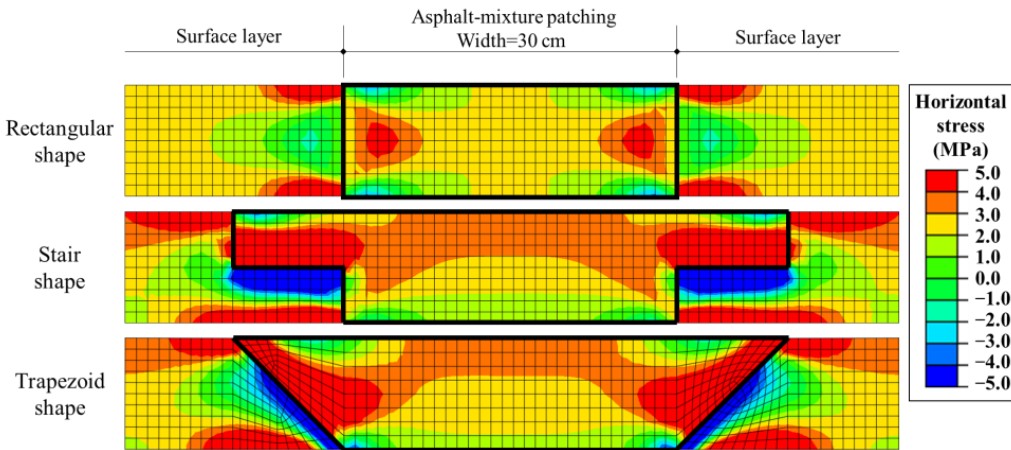

**Figure 11.** Horizontal-stress distribution during cooling process.

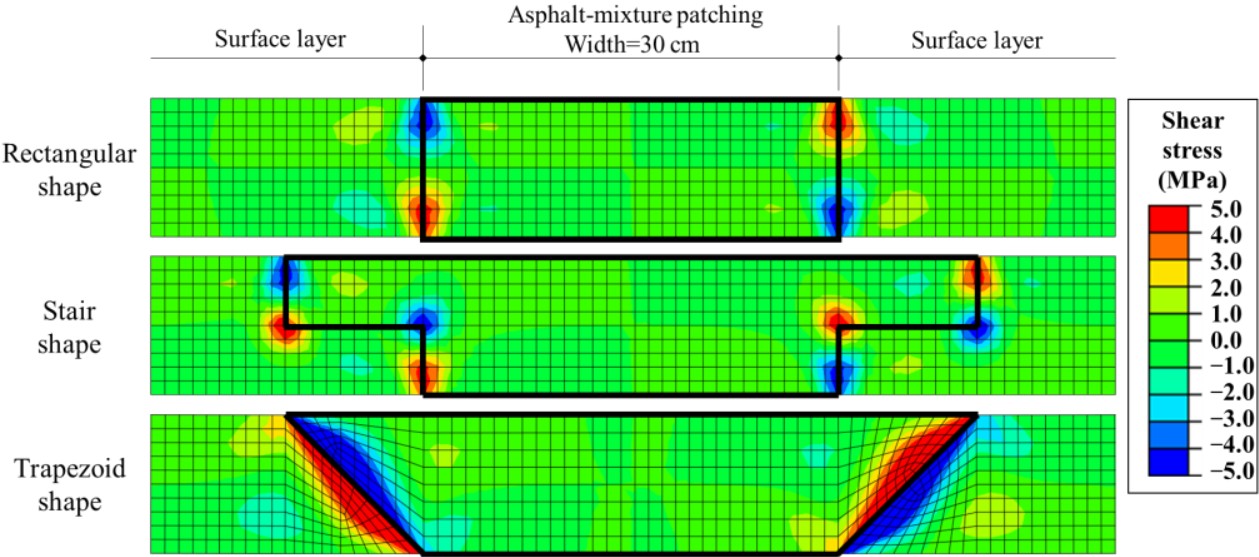

**Figure 12.** Shear-stress distribution during cooling process.

The locations of the maximum horizontal and shear stresses caused by the temperature decline were extracted and presented in Figure 13. It can be observed that the trapezoid-patched-asphalt pavement has the highest horizontal and shear stresses during the cooling process, followed by stair- and rectangular-patched pavements. This phenomenon can be explained by the observation that the temperature stress was closely related to the volume of the patched area. The rectangular shape requires the smallest patch area and hence causes the lowest stresses among the three pavements. According to the position of the peak value, the peak values of horizontal and shear stresses in the rectangular-patched pavement occur in the middle and surface of the asphalt layer. In the trapezoid-patched-asphalt pavement, the peak values of horizontal and shear stresses exist in the surface and middle of the asphalt-mixture layer, respectively. Thus, it can be deduced that the interface debonding can easily occur near the surface of the rectangular- and trapezoid-patched pavements. However, in the stair-patched pavement, the peak values of horizontal and shear stresses occur in the middle of the asphalt layer. Therefore, the stair shape was expected to reduce the interface debonding caused by the temperature stresses in the construction process.

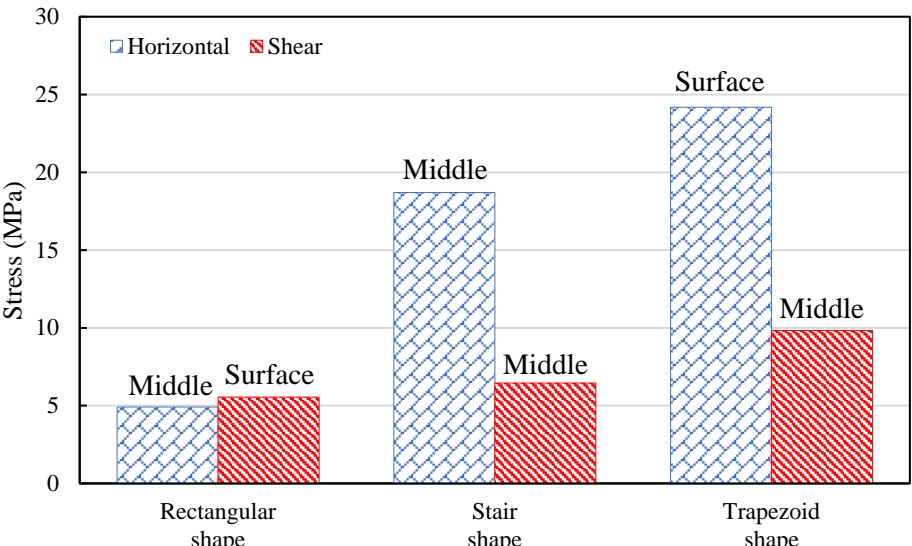

**Figure 13.** Maximum horizontal and shear stresses in the three patched-asphalt pavements.

### 3.4. Interface Debonding

The normal and tangential relative displacements between patched and unpatched parts in pavements were extracted based on the three-dimensional simulations. Figure 14 demonstrates the interfacial behaviors of the three patched parts in Configuration 1. A remarkable difference can be observed in the distributions of the relative displacements between patched and unpatched parts in the three model interfaces. In the normal direction, the top area of the rectangular-patched interface exhibits positive displacements, while the other two patched interfaces present positive displacement in the bottom areas. The tangential-relative-displacement distributions provide more information. Higher tangential relative displacements are distributed in the interface of the rectangular-patched model than in the other two models. The abovementioned phenomena indicate that the interface of the rectangular-patched pavement was the most vulnerable among the three pavements. For the interfaces of stair- and trapezoid-patched pavements, the distributions of the normal relative displacements are similar, while the interface of the trapezoid-patched pavement exhibit higher tangential relative displacements. Such difference can be explained by the continuity of the interfaces in the two pavements. For the trapezoid-patched pavement, the interface between patched and unpatched parts was continuous, within which the tangential relative displacements could easily be distributed. On the other hand, the interface in the stair-patched pavement was discontinuous, and the platform in the middle of the patched part effectively alleviated the tangential relative displacements. Therefore, the interface-debonding behavior in the trapezoid-patched pavement would be relatively more severe than in the stair-patched pavement.

The distributions of normal and tangential relative displacements in the interfaces of the three pavements in Configuration 2 are presented in Figure 15. View 1, View 2, and View 3 were obtained from different interfaces, and they suggest a relationship between the locations of the tire loading and the interface-debonding performances. A general conclusion that can be determined from the displacement distributions is that the interface of the rectangular-patched pavement was the most dangerous among the three pavements, which is in accordance with Configuration 1. Similarly, the interface of the trapezoid-patched pavement was more vulnerable than that of the stair-patched pavement due to the higher relative displacement in the continuous interface of the trapezoid-patched pavement. Moreover, for the trapezoid-patched pavement, the interface-debonding performances distant from the tire loadings (View 2 and View 3) were more severe than those near the tire loadings (View 1).

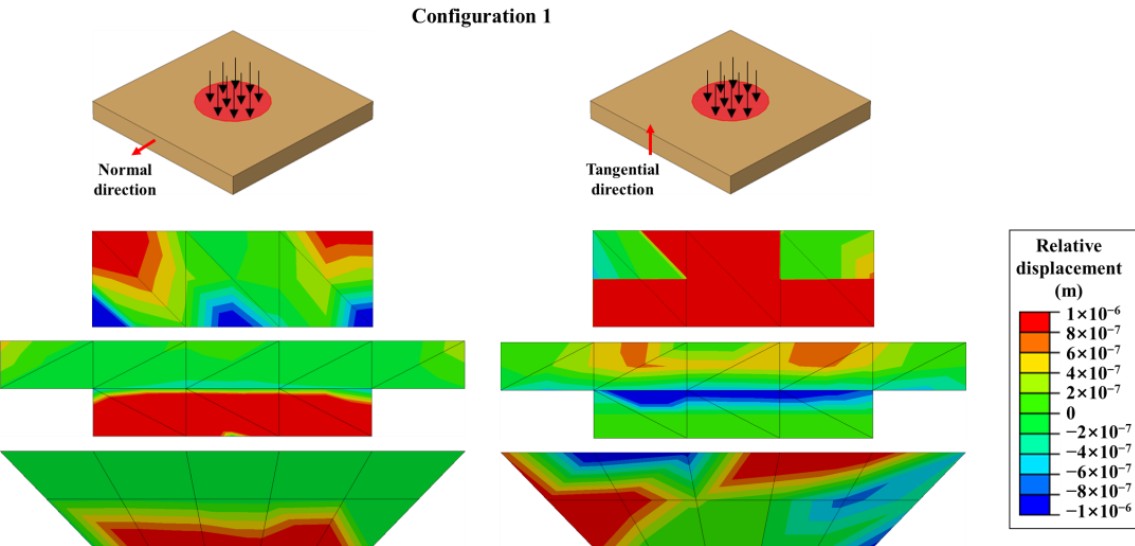

**Figure 14.** Interfacial behaviors of the three patched parts in Configuration 1.

The distributions of normal and tangential relative displacements in the interfaces of the three pavements in Configuration 3 are presented in Figure 16. View 1 and View 2 were obtained from different interfaces. The interface of the rectangular-patched pavement was the one most vulnerable to debonding deteriorations. In addition, the interface-debonding performances of the stair- and trapezoid-patched pavements were similar.

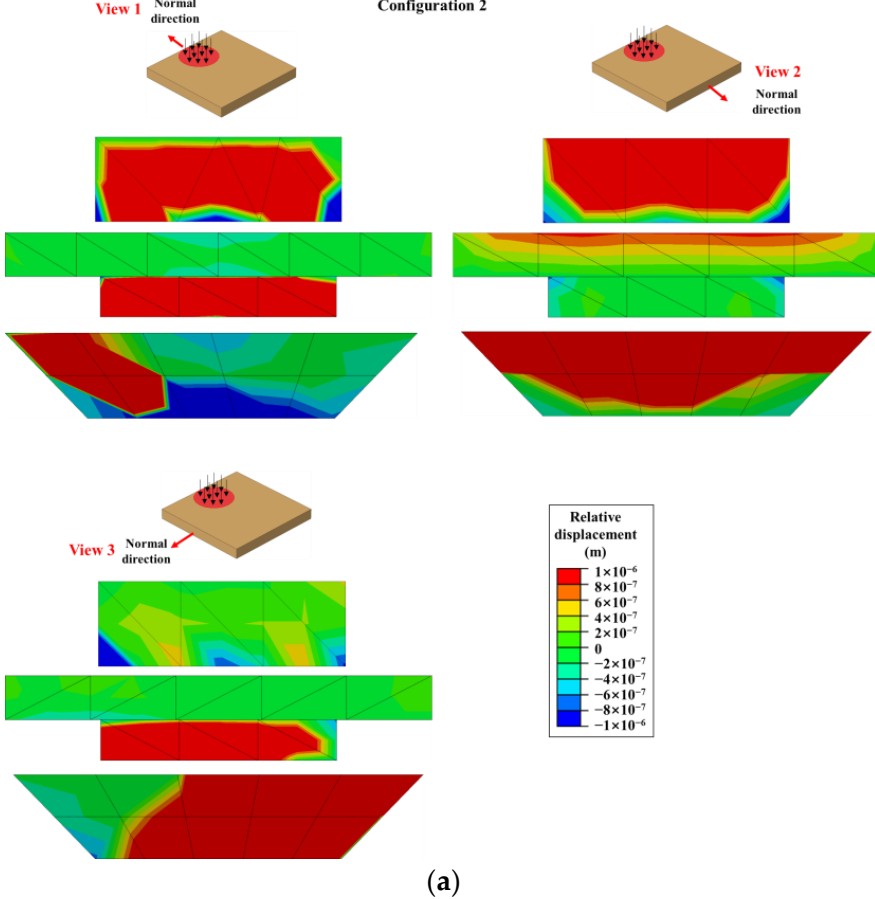

(**a**)

**Figure 15.** *Cont.*

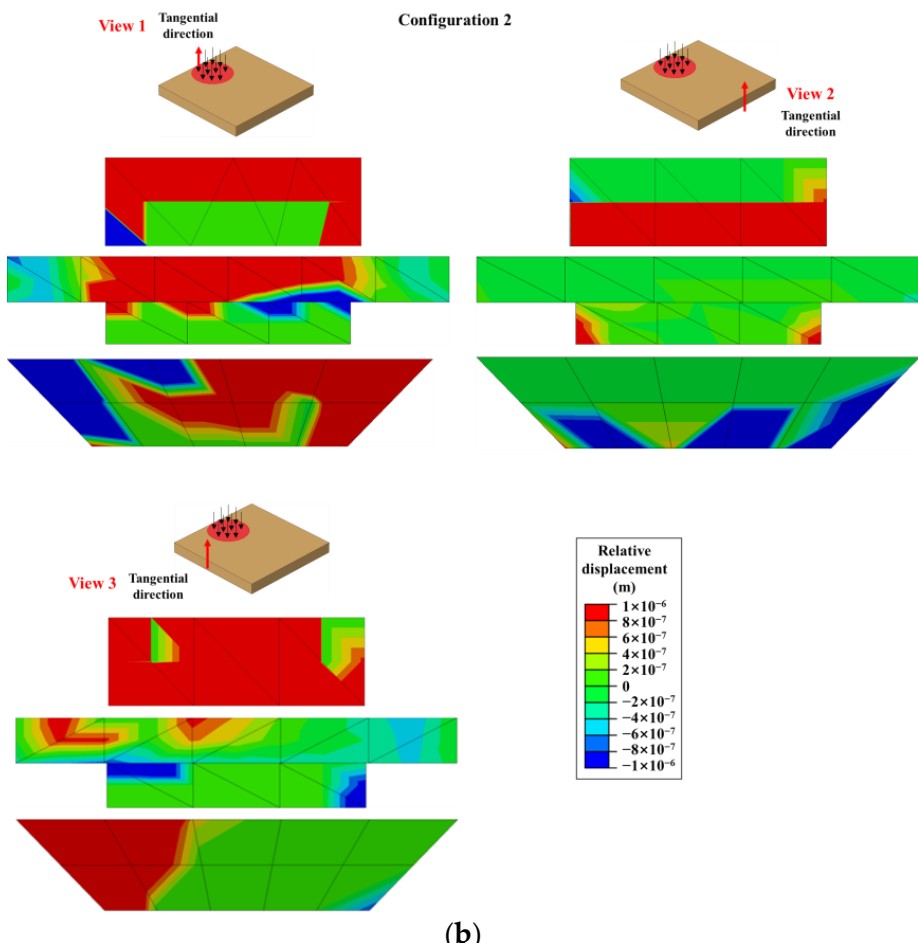

(**b**)

**Figure 15.** Interfacial behaviors of the three patched parts in Configuration 2 in the (**a**) normal direction and the (**b**) tangential direction.

To sum up, the interface-debonding performances of the three patched pavements with respect to the three-dimensional-stress states and tire-loading locations were effectively characterized. The rectangular-patched pavement was the most vulnerable to interface-debonding distress, followed by the trapezoid-patched pavement, and the stair-patched pavement showed the least interface-debonding behavior.

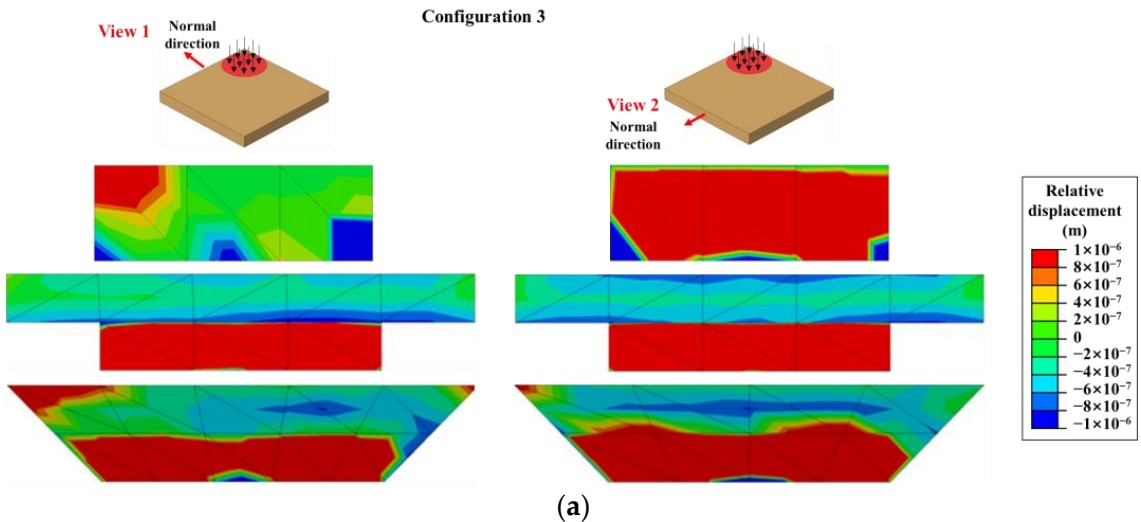

(**a**)

**Figure 16.** *Cont.*

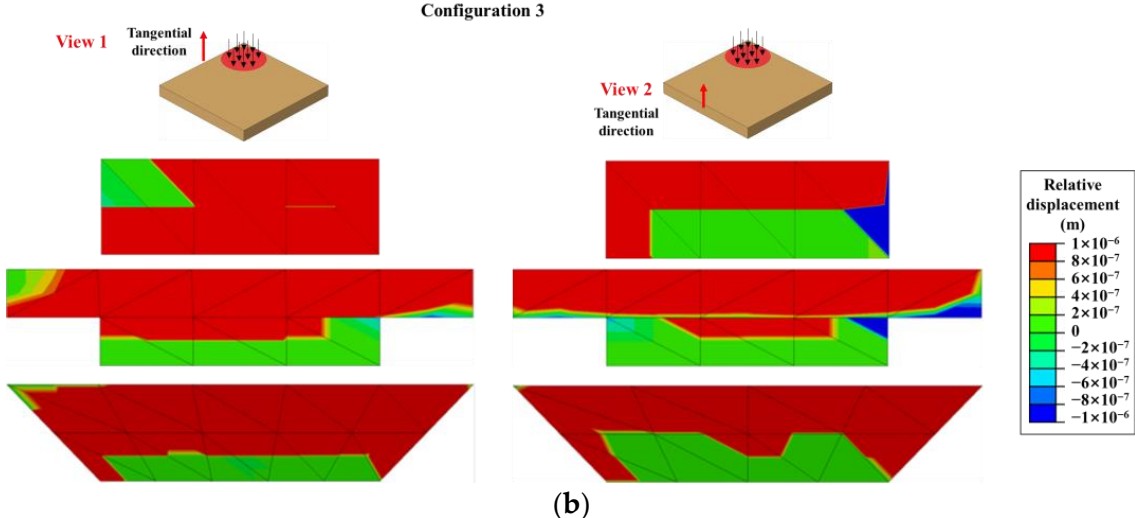

(**b**)

**Figure 16.** Interfacial behaviors of the three patched parts in Configuration 3 in the (**a**) normal direction and the (**b**) tangential direction.

## 4. Conclusions

To deeply investigate the mechanical performance of patched-asphalt pavements embedded with different patching shapes, systematic finite element simulations were conducted in this study. The finite element models of three patching shapes, namely rectangular, stair, and trapezoid patches, were established and simulated based on two- and three-dimensional simulations. In the two-dimensional simulation, the interface between patched and unpatched parts was perfectly bonded, and the crack propagation throughout the patched- and unpatched-asphalt mixture in the pavement was examined by the extended finite element method (XFEM). Consequently, performances of Top-Down cracks and Bottom-Up cracks were clearly illustrated, and possible explanations for crack propagation were discussed according to strain distributions. Further, the temperature stress caused by the patched-asphalt mixture in the pavements was analyzed using two-dimensional simulations. In the three-dimensional simulation, the interface-debonding performance with respect to three-dimensional-stress state and tire-loading locations were effectively characterized. The main conclusions are as follows.

(1). The rectangular-patched pavement was the type most vulnerable to Top-Down and Bottom-Up crack propagation, which relates to the vertical interface between patched and unpatched parts in the pavement.

(2). The stair- and trapezoid-patched pavements can effectively reduce the crack performance due to the transition of the patch parts. For trapezoid-patched pavement, the slant interface between the patched and unpatched part has a better ability to reduce the stress and strain discontinuities in the interface and to reduce the cracks in pavements, especially for Top-Down cracks.

(3). The slant interface of the trapezoid-patched pavement effectively alleviated the strain discontinuities between patched- and unpatched-asphalt mixtures, reducing Top-Down and Bottom-Up crack propagation. However, significant stress concentration was induced in the Bottom-Up crack tip in the trapezoid-patched pavement, which indicates that the Bottom-Up crack can further propagate in the trapezoid-patched pavement.

(4). Higher temperature stress is distributed in the stair- and trapezoid-patched pavements, which can be ascribed to the larger volume and interface of the two patched parts. In addition, the temperature stress near the interface of the trapezoid-patched pavement was higher than that in the stair-patched pavement, which indicates that interface debonding was more likely to occur in the trapezoid-patched pavement during the construction process.

(5). The three-dimensional simulation demonstrated that the rectangular-patched pavement was the type most vulnerable to interface-debonding distress, followed by the trapezoid-patched pavement, and the stair-patched pavement showed the least interface-debonding behavior.

Based on the simulation results, the trapezoid patching is recommended for use in repairing asphalt-pavement potholes to improve the workability of patched-asphalt mixtures. However, there is still a lack of field tests that validate the simulation results. To reduce the modeling efforts, the trapezoidal shape has 45-degree-sloped sides in this study. In practice, engineers should select proper angles for trapezoidal-sloped sides based on engineering requirements. In addition, without deep investigations on the mechanical and chemical properties of materials, the detailed interfacial behavior between patched- and unpatched-asphalt mixtures are still unclear. For instance, the trapezoid patching shows interfacial debonding potential, despite its great crack-resisting capability due to its slanted interface. Therefore, future research work will focus on preparing in situ experiments to validate the mechanical and damage-related performance of the patched pavements, concentrating on the bonding performance of the interfaces between patched and unpatched parts.

**Author Contributions:** Conceptualization, C.D. and Y.T.; methodology, H.Z.; software, Z.W.; validation, X.Y.; formal analysis, S.W.; investigation, S.W.; resources, C.D.; data curation, Y.T.; writing—original draft preparation, S.W.; writing—review and editing, C.D.; visualization, H.Z.; project administration, C.D. and Y.T.; funding acquisition, S.W. All authors have read and agreed to the published version of the manuscript.

**Funding:** The research reported in this paper was supported by the National Key Research and Development Program of China (Grant No. 2022YFB2602102), the Natural Science Foundation of Shandong Province (Grant No. ZR2023QE185, ZR2023QE296), and the Natural Science Foundation of Jiangsu Province (Grant No. BK20230256). The authors gratefully acknowledge this financial support.

**Institutional Review Board Statement:** Not applicable.

**Informed Consent Statement:** Not applicable.

**Data Availability Statement:** Dataset available on request from the authors.

**Conflicts of Interest:** Shujian Wang was employed by Shandong Hi-Speed Construction Management Group Co., Ltd., while Han Zhang, Zijian Wang, and Xinpeng Yao were employed by Shandong Hi-Speed Group Co., Ltd.. The remaining authors declare that the research was conducted in the absence of any com-mercial or financial relationships that could be construed as a potential conflict of interest.

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
