# Peer review of "Mechanical Performance of Patched Pavements with Different Patching Shapes Based on 2D and 3D Finite Element Simulations"

_infrastructures, doi:10.3390/infrastructures9030061_

Round 1

Reviewer 1 Report (Previous Reviewer 4)

Comments and Suggestions for Authors

After making changes and additions, the manuscript may be accepted, but it does not have much scientific significance.

Author Response

Comment: After making changes and additions, the manuscript may be accepted, but it does not have much scientific significance.

Response: The authors are appreciated to your comment.

Reviewer 2 Report (New Reviewer)

Comments and Suggestions for Authors

Very nice article- congrats!

Reviewer 3 Report (New Reviewer)

Comments and Suggestions for Authors

The paper presents a study aiming at assessing the mechanical performance of patched asphalt pavements, investigating various patching shapes through 2D and 3D finite element simulations (rectangular, stair and trapezoidal shapes). Within the 2D analysis, the crack propagations through across the patched and unpatched asphalt portions was also accounted. The 3D simulations were executed considering also different position of the tire loading the surface. Various aspects were investigated in the analysis (top-down and bottom up crack propagation mechanism, temperature stress (due to construction techniques) and interface bonding behaviour. It was found that the trapezoidal-shaped patch was recommended to mitigate mechanical problems and improve the workability of the repairing asphalt mixture.

In general, there are no specific remarks to be addressed about theoretical and procedural contents of the manuscript. Introduction is suitable to contextualize the argument and reports relevant literature references. Methodology is well explained. The results are goodly presented, commented and interpreted. There are no specific literature citations all along the results section, otherwise such fact could be conceived considering the innovative argument. The following short things could be improved before the manuscript publication.

1.   Please, perform a careful revision of citations, since some incongruencies seems to be present (e.g., reference number [12] is inserted in the reference list but is not cited in the text).

2.   Figure 3, Line 179: it is quite obvious that “AC” acronym is referred to “ASB” (Asphalt Stabilized Base). But, I think that it can be directly substituted everywhere with “ABS” for the sake of uniformity.

3.   It is not clear if the trapezoidal shape has 45-deg-sloped sides (it seems). Please, specify it.

4.   Please, check if all the numerical indications about FEM contours are reporting the unit of measurement

5. It would be appreciated to read some info about the simulation accuracy for demonstrating that results are significant (is it possible to report some indications/graphs about convergency/errors/residuals, etc.?).

Author Response

This manuscript is a resubmission of an earlier submission. The following is a list of the peer review reports and author responses from that submission.

Round 1

Reviewer 1 Report

Comments and Suggestions for Authors

In reality, potholes don’t have standardized shapes like rectangles, stairs, or trapezoids. How does this study relate to real-world applications?

How to model the interface between patch materials and asphalt pavement?

How to consider the size of patch in the modeling?

There is a limitation on the study as only one modulus is considered for patching material.

The authors inserted an initial crack in pavement surface to study top-down cracking. How large is the initially inserted crack, and does its width impact the results of the study on top-down cracking?

Figure 6: Why does the crack not initiate at the edge of the tire but at some distance away from it?

The authors concluded that “trapezoid shape is recommended in reducing top-down cracking propagation”. How does this relate to real-world application?

Figure 7(b): Why does the left show negative shear strain, while the right shows positive shear strain?

Also, there is a notch in the top mesh (rectangular shape), and a notch at the bottom of the mesh (unpatched surface). Why is that?

Why is there a need to insert a crack at the bottom to simulate the bottom-up cracking? You can show the tensile strain at the bottom of crack.

Section 3.3. How does horizontal and shear stress generate when the temperature cools from 150C to 30C? Please explain.

The authors analyzed both strain and stress. Why is there a need to show the horizontal stress and shear stress? How do they compare with the strength of the materials?

Comments on the Quality of English Language

The quality of English is generally ok. Can be improved. 

Reviewer 2 Report

Comments and Suggestions for Authors

A large number of relevant simulation studies have been carried out in the manuscript, but the reliability of pure simulation analysis is easy to cause doubts, and a large number of experiments need to be added to support the accuracy of the simulation. The overall manuscript is low in innovativeness and therefore the reviewers do not recommend it for publication.

Reviewer 3 Report

Comments and Suggestions for Authors

This paper is interesting and the methods for improving pothole repair durability is always of value. Overall I have few comments regarding the paper structure, methodology, results or conclusions. I do however suggest the conclusions section be better structured as it wafts a bit around the important parts and is not succinct enough. The main concern is the practical implementation of the proposed repair shapes. Most potholes, if correctly fixed, are saw cut and look like the rectangular shape the authors investigated. As for the step and trapezoid shapes, I have never seen these implemented in the western world and would be curious how this is practically achieved on site. This may however be a common approach in China. Potholes are often fixed quickly and cutting stair or trapezoid shapes seems very time consuming, if even possible. The stair shape requires a milling machine as a minimum for clean cuts and a common gang with saw cutters won't be able to achieve the required shapes. I am not sure how anyone will achieve the trapezoid shape without specialised equipment which most gangs working for local authorities won't have. To summarise, my concerns are as to the practical implementation and how this research can be applied in real life. If the authors can provide examples of how these shapes can be practically implemented and guidance on their construction it would be beneficial.

Comments on the Quality of English Language

The use of English is sufficient

Reviewer 4 Report

Comments and Suggestions for Authors

This paper employed finite element methods to deeply analyze the mechanical performance of patched asphalt pavements embedded with different patching shapes. Three patching shapes were considered in pavement pothole repairing based on the 2- and 3-dimensional finite element models. In the 2-dimensional models, the Top-Down and Bottom-Up crack propagations were simulated to assess the anti-damage performance of the patched pavements with different patching shapes. In addition, the thermal stress behaviors within patched asphalt pavements were simulated using the 2-dimensional model to analyze the performance of the patched pavements during the cooling process in construction. Besides, the interface debonding performances of the patched asphalt pavements were simulated using 3-dimensional models.

Positive remarks:

1. The authors analyzed an important problem related to road use during periods of variable temperatures. (Mainly the problem + - 0 o C).

2. A model allowing the analysis of strains and stresses in the damage area has been developed.

Critical remarks:

1. Lack of information about the mechanical and chemical properties of the materials joined together.

2. The analysis of forces that determine the effectiveness of the connection (mechanical and chemical) was omitted.

3. No reference to any experimental study.

Round 2

Reviewer 1 Report

Comments and Suggestions for Authors

No more comments. 

Comments on the Quality of English Language

Ok.